# Freak wave events in 2005–2021: statistics and analysis of favourable wave and wind conditions

**Ekaterina Didenkulova**[1,⚥]**, Ira Didenkulova**[2]**, and Igor Medvedev**[3]

[1]Faculty of Informatics, Mathematics and Computer Science, HSE University, Nizhny Novgorod 603155, Russia
[2]Department of Fluid Mechanics, University of Oslo, Oslo 0316, Norway
[3]Shirshov Institute of Oceanology, Russian Academy of Sciences, Moscow 117997, Russia
[⚥]*Invited contribution by Ekaterina Didenkulova, recipient of the EGU Nonlinear Processes in Geosciences Division Outstanding Early Career Scientists Award 2020.*

**Correspondence:** Ira Didenkulova (irinadi@math.uio.no)

**Abstract.** Freak or rogue waves are unexpectedly and abnormally large waves in seas and oceans, which can cause loss of human lives and damage to ships, oil platforms, and coastal structures. Evidence of such waves is widely spread around the globe. The present paper is devoted to analysis of the unified collection of freak wave events from different chronicles and catalogues from 2005 to 2021. The considered rogue waves are not measured in situ data, but their descriptions, which have been found in mass media sources and scientific articles, are the data used. All of them resulted in damage to ships or coastal/offshore structures and/or human losses. The collection accounts for 429 events. First, the analysis based on their characteristics taken from the descriptions of the events (including locations, water depth, damages) is carried out. Second, the analysis of wave parameters taken from the climate reanalysis ERA5 is performed. Thus, the most probable background wave parameters at the time of the freak event (including wind speed, gusts, significant wave height, maximum wave height, peak wave period, skewness, excess kurtosis, Benjamin–Feir instability (BFI), and wave spectral directional width) for each freak wave event are determined.

## 1 Introduction

Anomalously large waves in the ocean (the so-called rogue, freak, or killer waves) can be dangerous for vessels, including large cruise ships and small fishing boats, as well as oil and gas pipelines and platforms. They may destroy or damage the coastal constructions and can lead to fatal consequences for people spending time on the beach or fishing on the rocks. Rogue waves have become a subject of continuous scientific investigations for more than 2 decades, after their existence was proven by registering a New Year Wave at the Draupner platform in the North Sea on 1 January 1995 (Haver, 2003). The most common properties of freak waves are their unusually great wave height for a given sea state, short lifetime, and unexpected formation. The mathematical definition, which is used in oceanography, is that freak waves are waves whose height is at least twice greater than the significant wave height ($H_s$ or SWH), which is itself defined as an average of one-third of the highest waves in the record. It can also be defined as 4 times the standard deviation of the surface elevation. This definition is often used in spectral wave models (Massel, 1996; Kharif et al., 2009). The difference in magnitude between the two definitions is often only a few percent. It is believed that the formation of a rogue wave is the result of different physical factors working together. The main reasons which play a key role in the process of rogue wave appearance are the following linear mechanisms: dispersive focusing, which is a time–space localisation of wave train energy (Kharif and Pelinovsky, 2003; Pelinovsky et al., 2011; Fedele et al., 2016), geometrical focusing in basins of variable depth (Didenkulova and Pelinovsky, 2011; Benetazzo et al., 2017), wave–current interaction (Lavrenov, 1998; Onorato et al., 2011; Toffoli et al., 2015; Shrira and Slunyaev, 2014a, b), and random superpo-

sition of steep waves (Gemmrich and Cicon, 2022). Among the nonlinear mechanisms, the most significant are the modulational instability or Benjamin–Feir instability (Slunyaev et al., 2011; Ruban, 2007; Kharif and Touboul, 2010; Onorato et al., 2006), the interaction of coherent structures as solitons and breathers (Pelinovsky and Shurgalina, 2016; Slunyaev, 2019; Gelash and Agafontsev, 2018; Akhmediev et al., 2016; Didenkulova, 2019; Didenkulova, 2022), and the wave–wave and wave–coast interaction in shallow water (Didenkulova and Pelinovsky, 2011; Chakravarty and Kodama, 2014; Peterson et al., 2003). Variable wind and gust also contribute to the extreme wave formation (Pleskachevsky et al., 2012).

The study of the problem of freak waves requires a multifaceted approach, including development of analytical theories and carrying out numerical simulations and experimental measurements. In situ measurements play an important role in the investigation of the characteristics and frequencies of the appearance of rogue waves in nature. Such in situ wave measurements are carried out in different locations of the world ocean, for example (Didenkulova and Anderson, 2010; Mori et al., 2002; Stansell, 2004; Christou and Ewans, 2014; Häfner et al., 2021). However, their number and location of measurements are limited.

It became obvious that freak waves can occur at any water depth and almost everywhere in the world ocean. Thus, to get more information about them, catalogues of rogue waves started to be compiled. Some chronology of freak waves from the 16th century to the beginning of the 21st century was presented in Liu (2007) CE1 TS1. This catalogue includes a description of the most well-known or reliably reported freak wave encounters from open sources. The catalogues of recent accidents associated with freak waves also include information about weather conditions and wave parameters in the region (Didenkulova et al., 2006, 2022; Nikolkina and Didenkulova, 2011, 2012; Liu, 2014; Didenkulova, 2020). There are also catalogues of freak waves for specific locations, for example Ireland (O'Brien et al., 2013, 2018) or the USA (García-Medina et al., 2018).

In the present article, we unit and classify all freak wave accidents from the mentioned catalogues using additional information that appeared in the literature, unifying the selection criteria and data analysis. Section 2 is devoted to the overall statistics of freak wave events during the period from 2005 to 2021, based on their descriptions. All freak wave accidents are mapped and divided by the place of their occurrence, i.e. deep/shallow/coastal events. We also consider damages caused by these events. The final database is compiled according to a unified standard and is freely available on the Internet. In Sect. 3, we advance from the superficial description of freak events to the evaluation of the wave and wind conditions during event occurrence. Here, the quantitative parameters of freak waves, background waves and wind conditions, such as wind speed, gusts, significant wave height, maximum wave height, peak wave period, skewness, excess kurtosis, Benjamin–Feir instability (BFI) index, and

wave spectral directional width, extracted from the global atmospheric and ocean reanalysis ERA5 model are discussed and analysed. This part is principally new and gives a new understanding of the most probable conditions and mechanisms for freak wave formation. Conclusions are given at the end.

## 2  Statistics of freak wave accidents in 2005–2021

The whole list of analysed events, which can be considered freak waves, can be found at https://www.ipfran.ru/institute/structure/240605316/catalogue-of-rogue-waves (last access: 3 April 2023). Most of these events are picked up from the catalogues (Liu, 2007, 2014; Didenkulova et al., 2006, 2022; Nikolkina and Didenkulova, 2011, 2012; Didenkulova, 2020; O'Brien et al., 2013, 2018; García-Medina et al., 2018) and are supplemented by the missed cases and the latest freak wave accidents. Thus, the considered time period is from 2005 to 2021. In general, these events are not in situ measurements but are based on eyewitness reports taken from mass media sources, different chronicles and collections, and scientific articles. The browser search was carried out by keywords: freak wave, rogue wave, extreme wave, monster wave, killer wave, large wave, high wave, and similar words in French and Russian. Supplementary, shipwreck-themed websites have been checked (such as https://www.fleetmon.com/maritime-news/?category=incidents, last access: 3 April 2023; https://www.cruisemapper.com/accidents, last access: 3 April 2023; https://www.mlit.go.jp/jtsb/marrep.html, last access: 3 April 2023, etc.) We believe that we cover most of the major accidents, as they were reported worldwide. All of them more or less satisfy the image of a freak wave accident: unpredicted by the eyewitnesses and caused damage and/or human injuries or losses. The majority of descriptions are accompanied by remarks such as "all of a sudden a big wave hit the boat", "when the sudden waves swept away", "a freak wave suddenly came out of nowhere", "three freak waves had materialized from nowhere in rough but not formidable seas", etc. Moreover, some descriptions give the heights of the freak wave(s) and background waves, which help us to validate the definition of freak wave, whose height should be at least twice greater than the significant wave height $H_s$. In addition, the data from the global atmospheric and ocean reanalysis ERA5 (to be discussed in Sect. 3) are used to draw a connection between weather conditions in the area, specifically the significant wave height and the data from the eyewitness reports. Here we use the significant height of combined wind waves and swell ($H_s$) taken from the data of reanalysis, which is calculated as 4 times the square root of the integral over all directions and all frequencies of the two-dimensional wave spectrum. The event is added to the list if based on both the eyewitness report(s) and ERA5 data its description and characteristics support the freak wave formation.

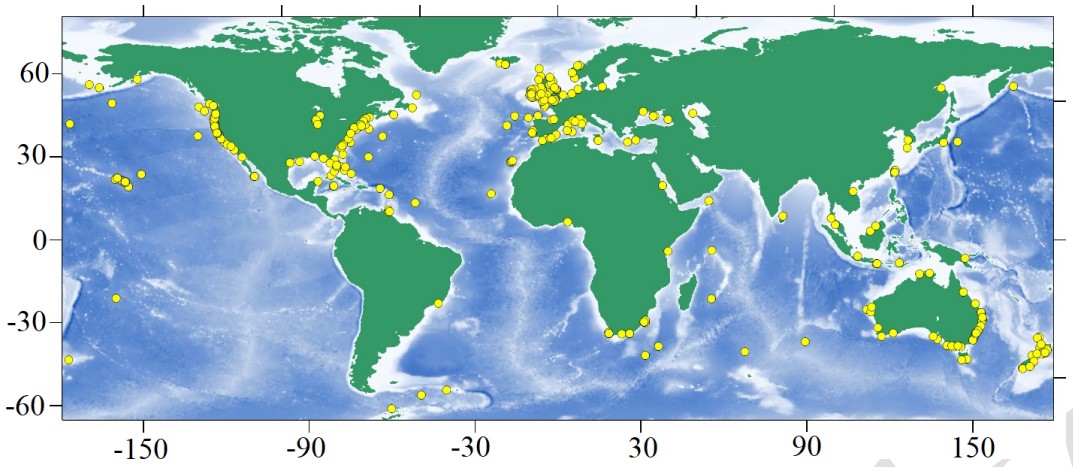

**Figure 1.** Map of freak wave accidents from 2005 to 2021.

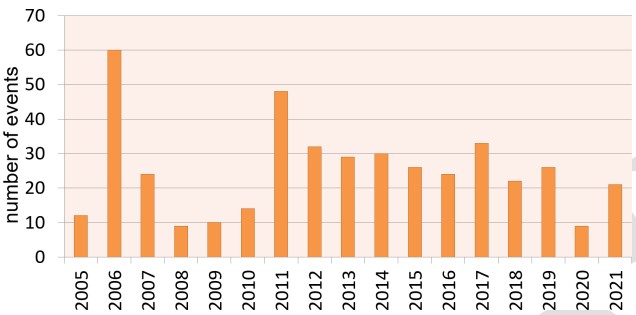

**Figure 2.** Distribution of freak wave accidents by years.

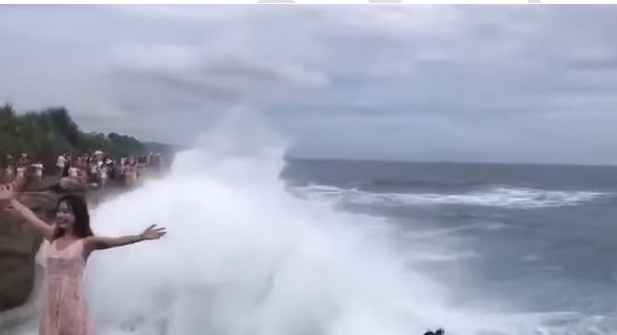

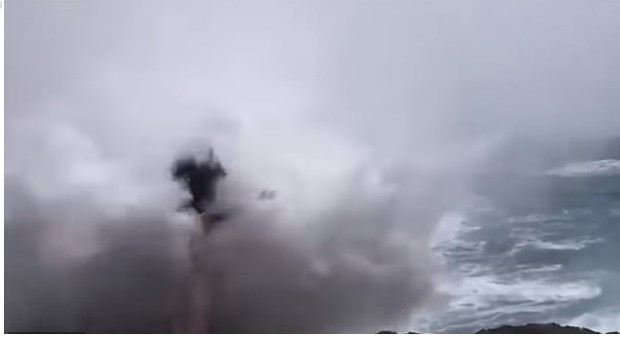

**Figure 3.** A person was almost swept out to sea from the cliff by a freak wave in Bali, Indonesia, on 17 March 2019 (@PDChinese).

The final list of events contains 429 freak wave accidents. Their locations are mapped in Fig. 1. It is clearly seen that their geography is widespread. The number of points increases closer to the coasts and water boarders because of
5 the more intensive use of these territories compared to the open ocean. The regions with the largest cluster of points are the east and west coasts of the USA and the coasts of Ireland and the United Kingdom, the Mediterranean Sea, and South Africa, as well as the southern and southeastern coasts
of Australia and New Zealand. Such distribution is governed by our search engine, as all mentioned territories are English-speaking regions. Although we have been limited to only three languages, the considered events show the widespread occurrence of freak waves in the world's oceans, the condi-
tions for their occurrence, and the damage they cause.

The distribution of freak wave accidents by years is presented in Fig. 2. It is not uniform, and deviations are significant. The year with the biggest number of freak waves from the list is 2006 (60 events). All of them happened in widely
spread geographic locations and were distributed evenly during the year. We assume that this year is associated with a public boom in freak waves, as many popular articles were published. After a while this topic became more "common",

and the amount of news on the topic decreased. In both 2008 and 2020 there were only nine events, which is the smallest 25 value in the histogram. The lack of events in 2020 can be explained by the restrictions during the COVID-19 pandemic, including a ban on visiting beaches in many countries.

Using the Multimaps service (https://multimaps.ru/, last access: 3 April 2023), the approximate depth of the events 30 is determined. A depth of 50 m is chosen to separate freak waves that occurred in deep areas from those in shallow areas. The threshold of 50 m has come from the characteris-

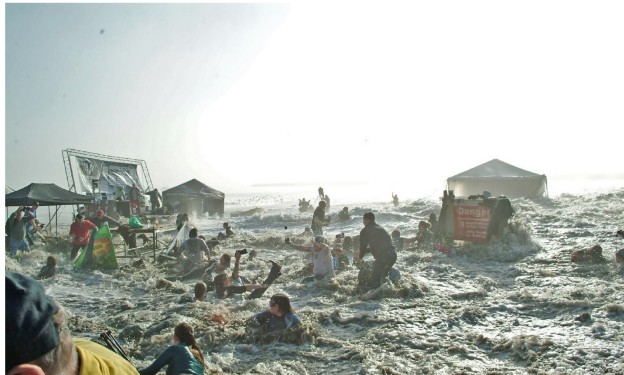

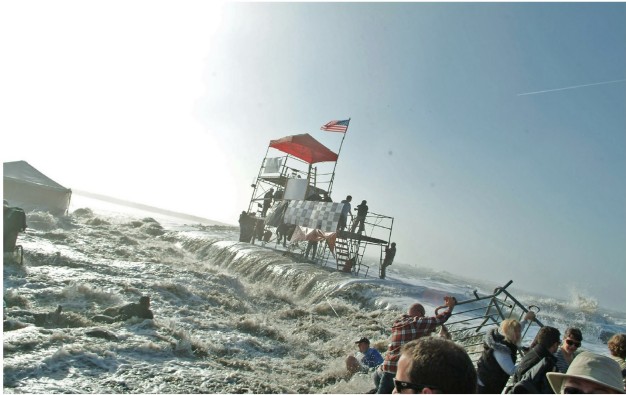

**Figure 4.** Freak wave accident on Mavericks Beach on 13 February 2010 (© Scott Anderson).

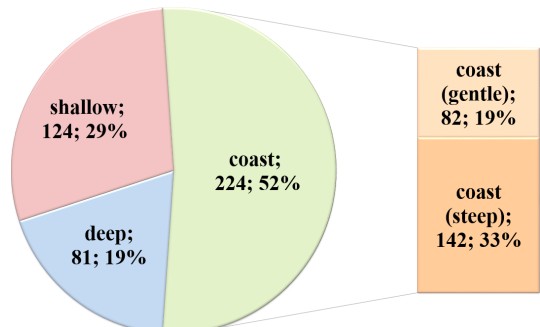

**Figure 5.** Distribution of deep, shallow, and coastal freak wave events.

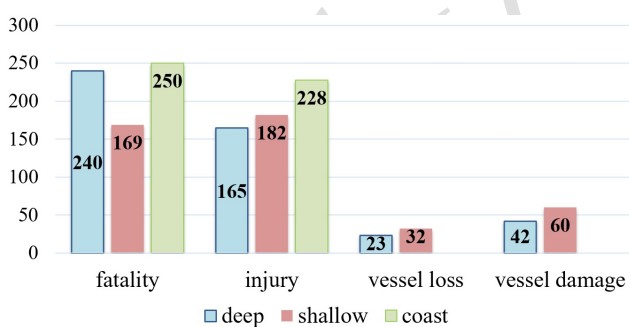

**Figure 6.** Damage caused by freak waves.

tic parameters for the North Sea, where deep waters are associated with water depths exceeding 50 m. There is also a class of events called coastal freak waves, which are divided into "gentle" (unexpected flooding on gentle beaches) and CE2 TS2 "rock" (unexpected surges on vertical constructions, i.e. rocks TS3 or embankments) freak events. Descriptions of several events of each mentioned type are given below. Figure 3 shows a freak wave event on the rocks. The person was almost swept away by a huge wave while posing for photos on a cliff in Bali (https://www.ibtimes.co.in/watch-bali-tourist-swept-away-by-huge-wave, last access: 3 April 2023). An example of a freak wave on a flat beach is shown in Fig. 4. A surfing competition took place on Mavericks Beach near San Francisco in California, USA. Two walls of water 1.8 m high took dozens of spectators by surprise, sweeping people off their feet. At least 13 people were seriously injured, including having broken legs and arms (https://www.thetimes.co.uk/article/rogue-waves-wipe-out-spectators, last access: 3 April 2023). One deep freak wave event was an accident involving the cruise ship *Louis Majesty*, when three freak waves smashed into the Mediterranean cruise ship. Two people were killed, and the cruise ship was affected by serious damages (https://www.youtube.com/watch?v=lvOceI6egg0, last access: 3 April 2023). An example of a shallow freak wave was an accident involving a whale-watching boat, named *Spirit of Gold Coast*, which was hit by a freak wave in Queensland (https://www.news.com.au/travel/travel-updates/incidents/, last access: 3 April 2023; https://www.youtube.com/watch?v=hWztpRKDmsg, last access: 3 April 2023).

The distribution of deep, shallow, and coastal freak wave events is shown in Fig. 5. There are 81 (19 %) events that occurred in deep areas; 124 (29 %) events in shallow areas; and 224 events (52 %) on the coast, including 82 (19 %) on the gentle beaches and 142 (33 %) on high cliffs and coastal walls. The number of freak wave observations on high cliffs and sea walls is significantly larger than on gentle beaches, which is in a good agreement with theoretical findings (Didenkulova and Pelinovsky, 2011).

One more criterion which unites all considered freak waves is the damage caused. The listed events led to human injuries (575) and deaths (658), as well as vessel damages (102) and losses (55), including small fishing boats and large ships (Fig. 6).

In spite of the larger number of shallow area events compared to those in the deep areas, the number of fatalities that occurred in deep areas is greater. Such a large number of human losses is also connected to two incidents. The first is an accident involving a fishing boat that sunk near Cape Inubōsaki on 23 June 2008 when 20 people drowned, and the second is the capsizing of the ferry *Rabaul Queen* on the

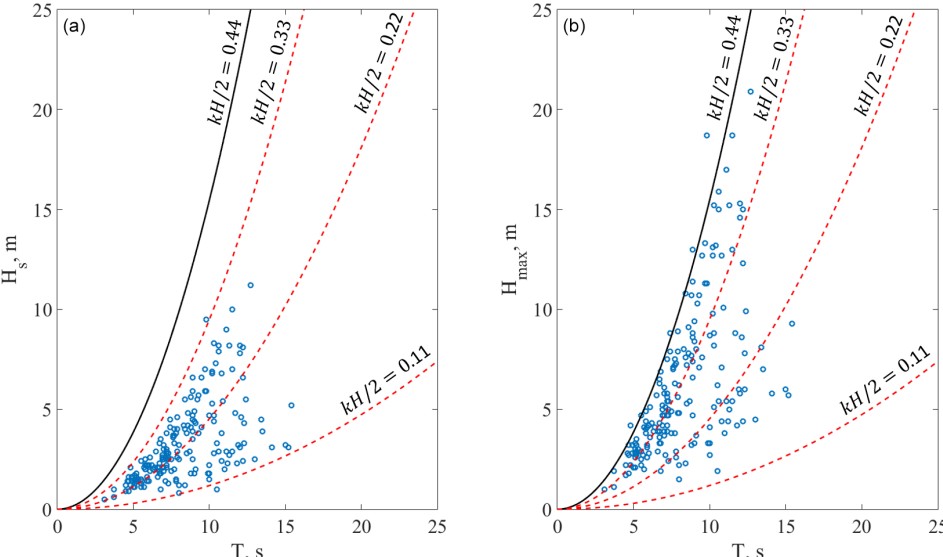

**Figure 7. (a)** Significant wave height versus wave period. **(b)** Individual maximum wave height versus wave period. Black line corresponds to the maximum steepness curve ($kH/2 = 0.44$).

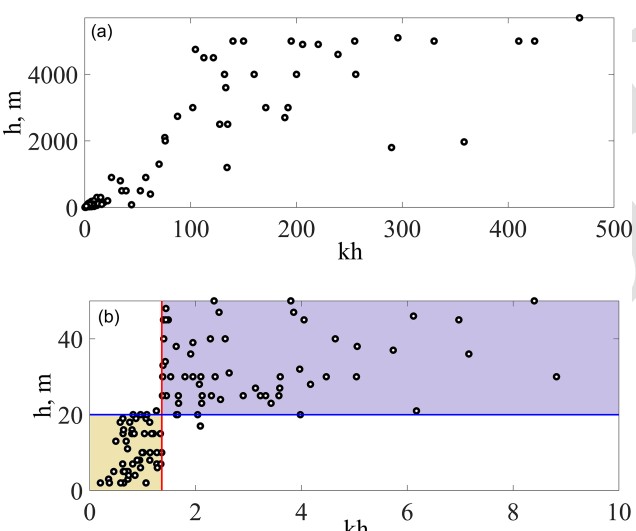

**Figure 8.** The parameter $kh$ versus the water depth (red line corresponds to the threshold of the criterion of modulational instability): **(b)** is a zoomed-in image of **(a)**.

east of Lae on 2 February 2012 when 126 people drowned. Among the coastal accidents the most dramatic is the one that happened on the west coast of South Korea when at least eight people were reported to have been killed after they were swept away by a 4–5 m high wave; at least 28 people were injured. During the freak accident, no specifics in meteorology were observed (Yoo et al., 2010).

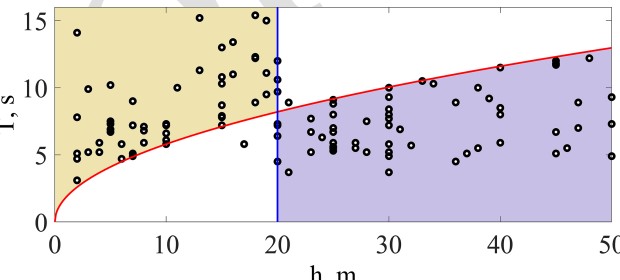

**Figure 9.** Period of freak waves plotted against the water depth of their occurrence; the solid red line corresponds to Eq. (3).

## 3  Analysis of freak wave characteristics based on atmospheric reanalysis ERA5

Apart from the freak wave parameters taken from the descriptions of the events and analysed in the previous section, in-depth analysis of the characteristics of background sea waves and wind has been performed using the data from the fifth generation of ECMWF atmospheric reanalysis of the global climate, ERA5 (Hersbach et al., 2020). The ERA5 reanalysis was developed using model cycle 41r2 of the 4D-Var data assimilation from the Integrated Forecast System (IFS). This reanalysis covers the period from 1979 to present. The characteristics of background waves, wind and freak waves have been determined, including wind speed, gusts, significant wave height, maximum individual wave height, peak wave period, skewness, excess kurtosis, BFI, and wave spectral directional width. These parameters were calculated from the two-dimensional wave spectrum, which includes both

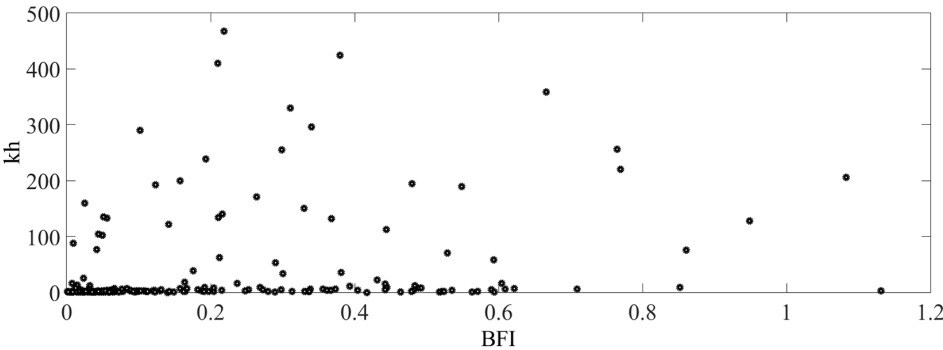

**Figure 10.** Benjamin–Feir instability (BFI) index versus the parameter $kh$ for deep and shallow events.

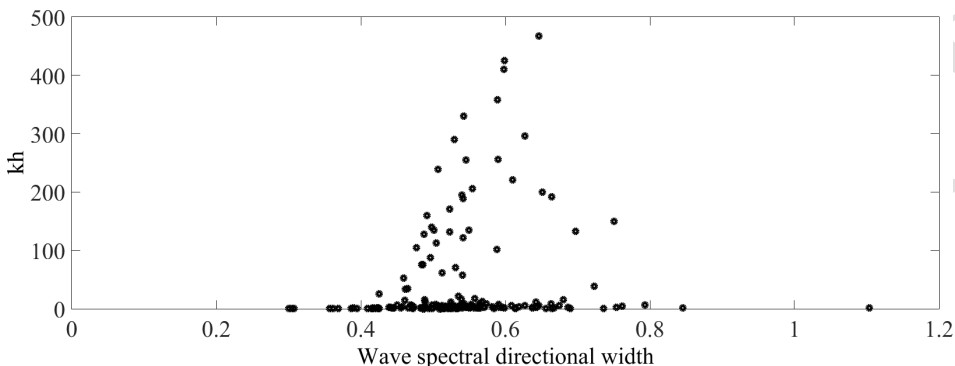

**Figure 11.** Wave spectral directional width versus the parameter $kh$ for deep and shallow events.

waves and swell. The most probable wind and wave conditions for freak wave generation have been discussed.

The maximum individual wave height ($H_{max}$) is an estimate of the greatest expected individual wave height within a 20 min time window, which is statistically derived from the two-dimensional wave spectrum. The wave spectrum can be decomposed into wind–sea waves, which are directly affected by local winds and swell, the waves that were generated by the wind at a different location and time. This parameter takes both into account. It can be used as a guide to the likelihood of extreme or freak wave occurrence. If the maximum individual wave height is more than twice the significant wave height, the corresponding 20 min interval may contain at least one freak wave, and the considered wave can be regarded as freak. In our dataset the estimated ratios $H_{max}/H_s$ mostly belong to the range from 1.8 to 2. Accepting the error in the 10 %, we can assert that analysed events fulfil the amplitude criterion of freak waves (Kharif et al., 2009). One of the reasons for this error is that $H_{fr}$ (freak wave height) is often unknown, while $H_{max}$ is statistically derived from the two-dimensional wave spectrum. It can be considered close to $H_{fr}$ but with a certain error, which we set as 10 %. Of course, this approach is not very accurate, since we are not talking about in situ measurements.

According to data of reanalysis from ERA5, the significant wave heights from the database ranged from 0.5 to 11.2 m,

the peak period ranged from 3.1 to 15.4 s, and the maximum individual wave height ($H_{max}$) ranged from 1 to 20.9 m.

The sea state steepness can be analysed by plotting the significant wave height against the peak period (Christou and Ewans, 2014). Figure 7 plots the significant wave heights against peak wave periods (a) and individual maximal wave heights against peak wave periods (b) for each freak wave event. The black line corresponds to the maximum steepness of Stokes' wave $kH/2 = 0.44$ ($k$ is the wave number, $H$ is the wave height) after which the irreversible process of wave breaking begins (Toffoli et al., 2010). However, individual waves can break well below the steepness 0.44. Indeed, sea states with a characteristic steepness of 0.12 have frequent wave breaking. For this reason, we also plot several lines corresponding to different steepnesses ($kH/2 = 0.44$, $kH/2 = 0.33$, $kH/2 = 0.22$, $kH/2 = 0.11$). The cloud of dots formed by maximum wave heights is clustered more toward the curve of maximum steepness. However, a large part of the cluster falls within the dots of $H_s$ from the first plot. Thus, the wave steepness cannot be the single factor in a freak wave event (Christou and Ewans, 2014).

One of the most important questions concerning freak waves is the reason for their appearance. Nowadays it is believed that modulation instability is the main mechanism of freak wave formation in the deep-water regions (Benjamin and Feir, 1967; Onorato et al., 2001; Dyachenko and Za-

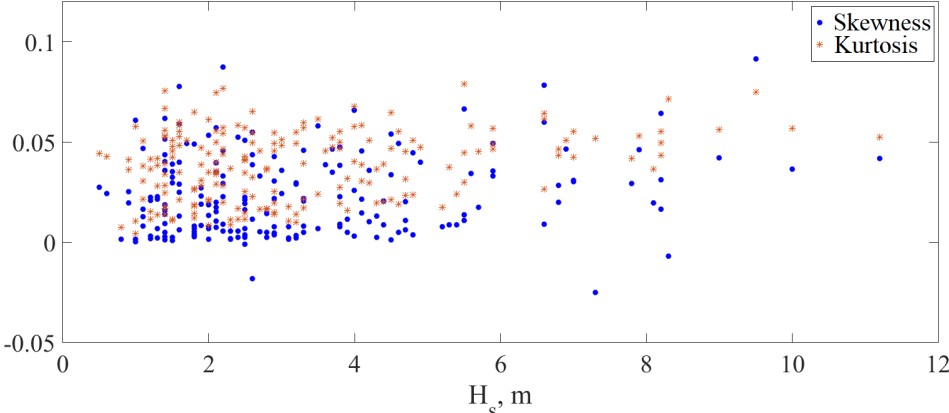

**Figure 12.** Distributions of skewness and excess kurtosis versus significant wave height.

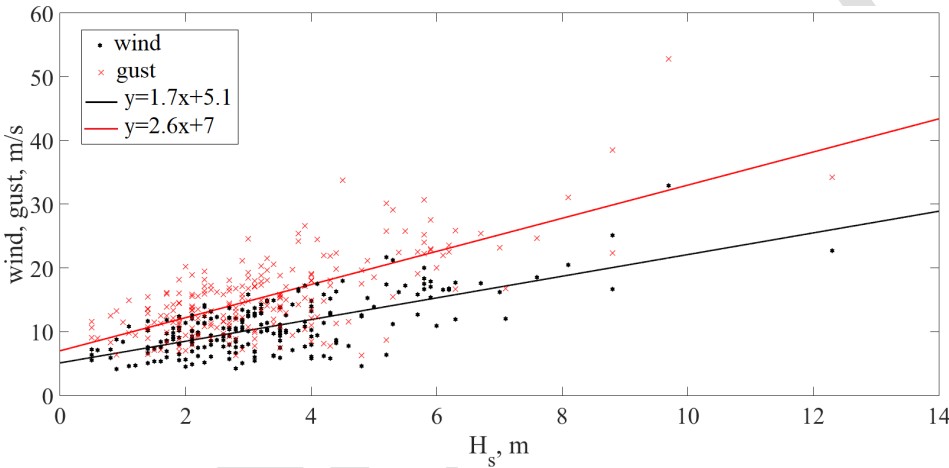

**Figure 13.** Dependence of wind speed and gusts on significant wave heights for coastal freak wave events.

kharov, 2005). However, closer to the coast, the role of modulational instability should be diminished (Kharif et al., 2009), and other mechanisms such as dispersive focusing (Fedele et al., 2016), geometrical focusing, or wave–current interactions should be prevalent. Using data obtained from the reanalysis model ERA5, we have checked if chosen freak events satisfy the criterion of modulation instability:

$$kh > 1.363, \tag{1}$$

where $h$ is the water depth, and $k$ is the carrier wave number (Osborne, 2010).

The approximate coordinates of the event were determined according to the reports of eyewitnesses. The corresponding depths were obtained using the Multimaps service (https://multimaps.ru/, last access: 3 April 2023).

Further, we can use the dispersion relation for gravity waves

$$\omega = \sqrt{gk \tanh{(kh)}}, \tag{2}$$

where $\omega = 2\pi/T$ is the angular wave frequency, and $T$ is the period. Wave periods are estimated using reanalysis data. Thus, $k$ can easily be found from Eq. (2).

The parameter $kh$ versus $h$ is plotted in Fig. 8a. However, it is more informative to look at the region of intermediate water depth (Fig. 8b). Points located to the right from the red line correspond to modulationally unstable waves. Almost all of these events occurred at the water depth greater than 20 m. Contrariwise, points located to the left from the red line are stable waves, and the depth of these events does not exceed 20 m. Despite the fact that the coordinates and depths of the freak wave events were determined approximately, a depth of 20 m can be chosen as a critical water depth that separates stable and unstable wave regimes. Thus, the criterion of modulation instability is well applied for water depth greater than 20 m according to the considered data of freak wave events. This conclusion coincides with the one made by Didenkulova et al. (2013), who used a small number of data.

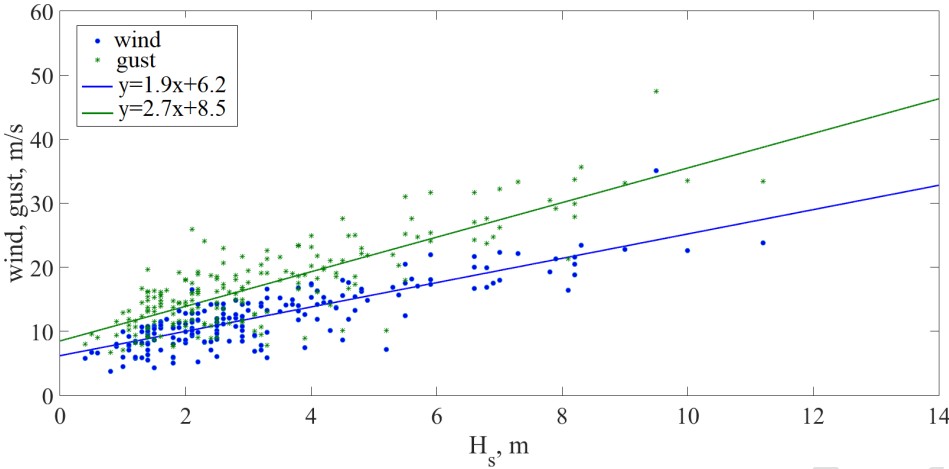

**Figure 14.** Dependence of wind speed and gusts on significant wave heights for deep and shallow freak wave events.

The modulational instability criterion can also be rewritten using the wave period $T$ and the water depth $h$:

$$T \le \sqrt{\frac{4\pi^2 h}{a_0 g}}, \tag{3}$$

where coefficient $a_0 \approx 1.195$ is taken from the approxima-
5 tion formula for the wave number in Hunt et al. (1979). Plotting the dependence of the wave periods versus water depths (Fig. 9), we obtain the same results as above (only intermediate depths are considered here). The 20 m water depth separates the modulationally stable and unstable waves quite ac-
10 curately. The red line in the figure corresponds to Eq. (3).

Another parameter that determines the fulfilment of the modulation instability conditions and is based on the wave spectrum is the Benjamin–Feir instability index (BFI). BFI is proportional to the ratio of two dimensionless parameters:
wave steepness and the spectral bandwidth. For the wave instability to occur, the condition BFI > 1 must be satisfied. The BFI index with an application to the real sea states was discussed in Alber (1978). Typical marine spectra turned out to be modulationally stable; therefore, the effect of self-
modulation of surface waves in real sea states for many years was considered minor. The BFI parameter was "reopened" for real sea waves in the very beginning of the 2000s; however, the application of the BFI index still faces difficulties: (i) the procedures for its calculation are very sensitive
to small changes in the input data, and (ii) the resulting maps of large BFI values generally poorly correlate with direct measurements of extreme waves by buoys (see for example Azevedo et al., 2022).

We have extracted the BFI data from the ERA5 reanaly-
30 sis model (see Fig. 10), but these data (which are averaged in some sense) are difficult to use for the considered freak wave events.

We have also looked at the wave spectral directional width extracted from ERA5 (Fig. 11). This parameter indicates

whether waves (generated by local winds and associated with 35 swell) are coming from similar directions or from a wide range of directions. The sea surface wave field consists of a combination of waves with different heights, lengths, and directions (known as the two-dimensional wave spectrum). Many ECMWF wave parameters (such as the mean wave pe- 40 riod) give information averaged over all wave frequencies and directions, so they do not give any information about the distribution of wave energy across frequencies and directions. This parameter gives more information about the nature of the two-dimensional wave spectrum and represents 45 a measure of the range of wave directions for each frequency integrated across the two-dimensional spectrum. It takes values between 0 and $\sqrt{2} \approx 1.4$, where 0 corresponds to a unidirectional spectrum and $\sqrt{2}$ indicates a uniform spectrum (i.e. all wave frequencies coming from a different direction). 50

According to Fig. 11, this parameter is mainly distributed between 0.4 and 0.7. This suggests that a crossing sea regime should not play a major role in the considered freak wave data.

Higher statistical moments have been analysed for deep 55 and shallow events. Skewness takes values between $-0.0251$ and 0.0913. Excess kurtosis takes values between 0.0041 and 0.0789. Their distributions versus significant wave height are presented in Fig. 12. This shows that a probability of freak wave occurrence is larger than for the Gaussian process. 60

It was previously noted that wind gusts may increase the local wave and freak wave heights (Touboul et al., 2006; Pleskachevsky et al., 2012). Using the reanalysis data, the winds and gusts for all considered freak wave events were estimated. Wind gust is the maximum wind gust at the speci- 65 fied time at a height of 10 m above the earth surface. It is defined as the maximum of the wind averaged over 3 s intervals. This duration is shorter than a model time step, and so the ECMWF Integrated Forecasting System (IFS) deduces the magnitude of a gust within each time step from the time-step- 70

averaged surface stress, surface friction, wind shear, and stability. Care should be taken when comparing model parameters with observations because observations are often local to a particular point in space and time, rather than representing averages over a model grid box. Wind speed and gusts versus significant wave heights for coastal freak wave events and their linear approximations are presented in Fig. 13. The coefficients of determination for both wind speed and gust data for coastal events are around 0.5. In general, higher wind speeds and gusts generate greater wave heights. However, the standard deviation is essential for these distributions, and one can see from Fig. 13 that the same wind speed (for example $5\,\mathrm{m\,s^{-1}}$) can generate wave heights from 0.5 to 5 m. We should note that by having a resolution of approximately $1°$, the ERA5 model does not perform well in coastal areas with complicated bathymetry. Dependence of wind speed and gusts versus significant wave heights for shallow and deep freak wave events and their linear approximations are presented in Fig. 14. The coefficients of determination for both wind speed and gust data in this case are 0.68, which is larger than for coastal events. CE3 TS4

## 4 Conclusions

In the present article, the statistics of a united database of freak wave events reported in mass media sources and scientific literature from 2005 to 2021 are analysed. The database is freely available on the Internet and can be found at https://www.ipfran.ru/institute/structure/240605316/catalogue-of-rogue-waves (last access: 3 April 2023). The main source of information here is the eyewitness reports and not in situ measurements. It is shown that freak wave events are widely spread all over the world and lead to dramatic consequences for coastal structures, human lives, and navigation. The database includes 81 events (19 %) that occurred in deep areas (water depth more than 50 m); 124 (29 %) in shallow areas (water depth less than 50 m); and 224 events (52 %) on the coast, including 82 (19 %) on gentle beaches and 142 (33 %) on high cliffs and vertical structures. Events from the combined catalogue from 2005 to 2021 caused significant damage: 575 people were injured, 658 people were killed, 102 ships were damaged, and 55 ships, both small fishing boats and large ships, sunk.

An analysis of the characteristics of wave and wind conditions for each freak event was performed using data from the ERA5 fifth-generation ECMWF atmospheric reanalysis of the global climate. According to the coordinates of events taken from the descriptions, the characteristics of background waves, wind and freak waves were determined, including wind speed, gusts, significant wave height, maximum individual wave height, peak wave period, skewness, excess kurtosis, BFI, and wave spectral directional width. The values of skewness and excess kurtosis of corresponding sea states showed a deviation from the Gaussian distribution and a larger probability of freak wave occurrence. It was also shown that in general stronger winds and gusts generate greater wave heights. However, the standard deviation is rather large for these distributions, and the same wind can generate a wide range of wave heights. Using CE4 TS5 the data obtained from the ERA5 reanalysis model, an analysis of the feasibility of the modulation instability criterion and the TS6 involvement of this mechanism in the formation of a specific freak wave CE5 TS7 was performed. It was shown that according to the considered data of freak wave events, the criterion of modulation instability is well applicable for depths greater than 20 m. CE6 TS8

*Data availability.* All collected catalogue freak wave data from 2005 to 2021 are available at https://www.ipfran.ru/institute/structure/240605316/catalogue-of-rogue-waves (last access: 3 April 2023).

*Author contributions.* ED and ID collected and analysed the data of freak wave events from mass media sources. IM provided climate reanalysis ERA5 data of selected freak waves. ED prepared the original draft of the manuscript, which was reviewed and edited by ID and IM. All authors have read and agreed to the published version of the paper.

*Competing interests.* At least one of the (co-)authors is a member of the editorial board of *Natural Hazards and Earth System Sciences*. The peer-review process was guided by an independent editor, and the authors also have no other competing interests to declare.

*Acknowledgements.* The authors thank Mauricio Gonzalez and the two anonymous referees, whose critical comments helped to improve the manuscript. The authors are also grateful to Efim Pelinovsky and Alexey Slunyaev for fruitful discussions.

*Financial support.* This research has been supported by the Russian Science Foundation (grant no. 21-77-00003). Publisher's note: the article processing charges for this publication were not paid by a Russian or Belarusian institution.

*Review statement.* This paper was edited by Mauricio Gonzalez and reviewed by two anonymous referees.

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

**Remarks from the language copy-editor**

CE1    Please note the adjustment: only the year is in parentheses now.

CE2    Please give an explanation of why this sentence needs to be changed. We have to ask the handling editor for approval. Thanks.

CE3    Please give an explanation of why this needs to be changed. We have to ask the handling editor for approval. Thanks.

CE4    Please give an explanation of why this needs to be changed. We have to ask the handling editor for approval. Thanks.

CE5    Please give an explanation of why this needs to be changed. We have to ask the handling editor for approval. Thanks.

CE6    Please give an explanation of why this needs to be changed. We have to ask the handling editor for approval. Thanks.

**Remarks from the typesetter**

TS1    **No correction necessary.**

TS2    **Please insert changes.**

TS3    **Please insert changes.**

TS4    **Please insert changes.**

TS5    **Please insert changes.**

TS6    **Please insert changes.**

TS7    **Please insert changes.**

TS8    **Please insert changes.**