# Peer review of "Freak wave events in 2005-2021: statistics and analysis of favourable wave and wind conditions"

_Natural Hazards and Earth System Sciences, 2022_

## Author Response (AR1)

*We thank both referee for useful comments and suggestions. Below we give the point-by-point response to the referees' comments and indicate the changes made in the manuscript.*

**Response to Referee 1**

1)      The denomination of deep/shallow/coastal freak waves is a little bit misleading. Indeed, the classification is only based on the location of the rogue wave (whether it happened in waters of depth greater or smaller than 50m). This denomination is a little misleading, since it has no direct connection to the classical kh (dispersive) parameter. Maybe using "deep area" and "shallow area", or something similar, would facilitate comprehension of the reader.

*Answer: We agree. "Deep water" and "shallow water" are now replaced by "deep area" and "shallow area".*

2)      The beginning of section 3 would probably benefit from a more detailed description of the ERA5 data, and their processing. For example, I could not understand how the values of Hfr are obtained (Although it is pretty clear for Hs). The same remark can be made for the values of the gustiness. Yet, the findings in figures 7 to 12 are pretty good.

*Answer: In fact, Hfr was substituted by Hmax since Hfr which is the freak wave height cannot be obtained from the ERA5 data. Here Hmax (the maximum individual wave height) is the estimation of the highest waves according to the reanalysis.*

*We have also added the following clarifications in the text:*

*Line 148: "The maximum individual wave height ($H_{max}$) is an estimate of the expected largest individual wave height within a 20 minute time window, which is derived statistically from the two-dimensional wave spectrum. The wave spectrum can be decomposed into wind-sea waves, which are directly affected by local winds, and swell, the waves that were generated by the wind at a different location and time. This parameter takes account of both."*

*Line 228: "Wind gust is the maximum wind gust at the specified time, at a height of ten meters above the Earth surface. It is defined as the maximum of the wind averaged over 3 second intervals. This duration is shorter than a model time step, and so the ECMWF Integrated Forecasting System (IFS) deduces the magnitude of a gust within each time step from the time-step-averaged surface stress, surface friction, wind shear and stability. Care should be taken when comparing model parameters with observations, because observations are often local to a particular point in space and time, rather than representing averages over a model grid box."*

**Response to Referee 2**

*We thank the referee for the effort and valuable remarks. Below we highlight the scientific importance of our work and give the point-by-point response to the referee's comments.*

The database is only a marginal improvement compared to previously published literature. A similar database by the same group authors covers events up to 2018, I therefore don't think is worthed dedicating a full manuscript to this slightly extended period. I feel that maintaining a public accessible database would be more valuable.

*Answer: The new result here is the analysis of wave and wind conditions corresponding to each freak wave event using the reanalysis model ERA5, and nothing similar had been done before. Thus, we advance from the superficial description of freak events to the evaluation of the wave and wind conditions of the event. Moreover, the importance of this work is also in the unification of freak wave events from catalogues of different authors supplemented by missed cases; the database is compiled according to a unified standard and is freely available on the Internet for its convenient usage. This clarification is added in Line 61.*

How the database is compiled is not described/detailed. At line 86 authors mention "search engine", it would have been more helpful making the search engine publicly available, so that the output can be used and improved. However, no details on how the data mining process is conducted has been presented in the manuscript.

*Answer: The browser search was carried out by keywords in English, Russian and French, the languages which the authors speak. The collection includes a few cases from other languages, which were shared with us by our international friends and colleagues. Supplementary, shipwreck themed websites have been checked (such as https://www.fleetmon.com/maritime-news/?category=incidents, https://www.cruisemapper.com/accidents, https://www.mlit.go.jp/jtsb/marrep.html, etc.) This clarification is added to the text (Line 75).*

The analysis is poor, both in terms of statistical analysis of the dataset and possible biases in the search engine (i.e. how the dataset might be skewed towards "English-speaking" media, and how coastal areas and highly populated areas where more people might be likely to experience freak events) and the description of the results. For example the authors should have explored more the dataset For example the high number of events in 2006 that exceed the mean yearly value by a large margin is statistically significant? Are these events clustered in a particular region? Or associated with a particularly strong system?

*Answer: We agree with Referee's comments about incompleteness of this database. This is why our focus is not on freak wave geographical distribution, but freak wave accidents associated with damage and their impact on human lives and infrastructure. Since the considered data are based on the eyewitnesses' reports, it is not possible to take into account all events due to the lack of information. For example, the extreme wave was there, but it passed unnoticed since there was nobody at that specific location at that moment of time. This is why we say that in general the probability to observe freak wave accident in the coastal area is larger, since coastal areas are more populated. This basically means that there is a large number of eye-witnesses who can observe and experience the wave. Also, not all the events are reported in the mass media. Therefore, we do not put so much emphasis on the geographic distribution of the events and clusters, as referee suggests, but rather on types of beach slopes, where accidents took place, and background wave and wind conditions associated with the events. Our explanation about large number of freak wave events in 2006 is given in Line 100.*

I also think that the explanation of the physical mechanisms likely to lead to freak events is not scientifically sound and consistent. The authors identify 20m depth as a threshold for freak waves events. If results are used to inform the analysis, this value should have been used to separate deep water and coastal areas.

*Answer: Indeed, our usual classification is only based on the location of the rogue wave and the corresponding water depth, whether it is larger or smaller than the certain threshold, while wave period and consequently dispersive parameter kh is not known. The threshold of 50 m has*

*come from the characteristic parameters for the North Sea, where deep waters are associated with water depths exceeding 50 m. (Clarified in line 108).*

*On the other hand, the value of 20 m came as a result of this study by applying the criterion of modulational instability to the data of wave background from the ERA5 reanalysis model.*

*To make it clear we replaced "deep water" and "shallow water" by "deep area" and "shallow area".*

**Specific Comments**

In the abstract the authors claim that freak waves "occur without specific reasons", there is a very extensive research on the physical mechanisms that contribute to the formation of these events, and the authors later cite many of these works. The statement is therefore misleading.
*Answer: Agree. The phrase "occur without specific reasons" was removed from the abstract.*

The author states that all reported events resulted in ship or coastal/offshore structure damage and/or human losses. Isn't it obvious that events reported in the media are the ones that have had a high impact on human activities? And therefore only those that have caused damage and losses are reported. In summary there is a bias towards these events.
*Answer: We agree with the Referee's comment. However, it is not possible to analyze all freak waves happened in the World Ocean. We will always be limited by the registration system, its limitations in time and space. Therefore, we can only deal with those freak waves which were registered, either using satellites from space, or in-situ measurements, or using the descriptions of freak waves based on eye-witnesses reports. We use the last approach, and naturally it is almost always connected to some damages. That is why we can say, that we also study the impact of freak waves on human activity. So we do not see here any contradiction.*

I wonder why the search engine has been extended to include media in other languages. Chinese and Spanish are the languages most widely spoken in the world, I think inclusion of these would have greatly improved the general purpose of the manuscript.
*Answer: We underline that the list of analyzed rogue waves is incomplete. We use the browse search the languages we speak (English, Russian, French). Organizing the search in others languages is the task for a different field of science like neural networks, which is not the purpose of the present work. Nevertheless, the considered events show the widespread occurrence of freak waves in the world's oceans and show the conditions of their occurrence and damage they cause.*

The year 2006 was an exceptional year, I would have expected a more detailed analysis. The deviation from the long term mean seems statistically significant.
*Answer: Indeed, the 2006 year is a record holder of freak wave events. All of them happened in widely spread geographic locations and are distributed evenly during the year. We assume, that this year is associated with a public boom on freak waves, many popular articles were published. After a while this topic became more "usual" and the number of news decreased. This explanation is added in line 100.*

At the end of section 2 authors state that most of the fatalities are in the open ocean. A more detailed analysis of fatalities per accident would have been carried out. In other words, how many of the accidents resulted in fatalities would have been a better representation. As the authors hint, most of the fatalities are associated with single accidents that took many lives.
*Answer: Not in this way. We state that "In spite of the larger number of shallow area events compare to those in the deep areas, the number of fatalities happened in deep areas is greater than in the shallow ones." It does not mean that most of the fatalities happened in the open ocean. The*

*whole distribution of fatalities and injuries for deep, shallow water and coast is given in Fig. 6. It can be seen that most of fatalities happened in the coast (it is in agreement with the large number of accidents happened at the coast (Fig.5)). To clarify the importance of coastal events, the description of the most destructive coastal event has been added to the text in line 135: "Among the coastal accidents the most dramatic is one that happened on the west coast of South Korea when at least eight people are reported to have been killed after they were swept away by the high wave of 4–5 m; and at least 28 people were injured. For this freak accident, no special meteorological conditions were identified (Yoo et al., 2010)."*

Authors discuss modulational instability focussing on wave steepness alone, the Benjamin Feir Instability (BFI) is also affected by the shape of the spectrum (bandwidth in both frequency and directions).

The threshold $kh = 1.363$ the focussing to the defocussing regime in unidirectional sea state; it has been shown that crossing seas instead lead to the formation of rogue waves below the threshold (and directionality stabilises modulational instability in deep water).

*Answer: We thank the referee for these two comments. The following section has been added to the manuscript (Line 199):*

*«Another parameter that determines the fulfillment of the modulation instability conditions and is based on the wave spectrum, is the Benjamin Feir Instability index (BFI). BFI is proportional to the ratio of two dimensionless parameters: wave steepness and the spectral bandwidth. For the wave instability to occur, the condition BFI > 1 must be satisfied. The BFI index with an application to the real sea states was discussed in [Alber I.E. The effects of randomness on the instability of two-dimensional surface wavetrains. Proc. Roy. Soc. Lond. A. 1978; 363: 525–546]. Typical marine spectra turned out to be modulationally stable; therefore, the effect of self-modulation of surface waves in real sea states for many years was considered minor. The BFI parameter was "reopened" for real sea waves in the very beginning of the 2000s, however, the application of the BFI index still faces difficulties: (i) the procedures for its calculation are very sensitive to small changes in the input data, and (ii) the resulting maps of large BFI values generally poorly correlate with direct measurements of extreme waves by buoys [see for example, Azevedo, L.; Meyers, S.; Pleskachevsky, A.; Pereira, H.P.P.; Luther, M. Characterizing Rogue Waves in the Entrance of Tampa Bay (Florida, USA). J. Mar. Sci. Eng. 2022, 10, 507. https://doi.org/10.3390/ jmse10040507].*

*We have extracted the BFI data from the ERA5 reanalysis model (see Figure 10), but this data (which is averaged in some sense) is difficult to use for the considered freak wave events.*

*We have also looked at the wave spectral directional width extracted from ERA5 (Figure 11). This parameter indicates whether waves (generated by local winds and associated with swell) are coming from similar directions or from a wide range of directions. The sea surface wave field consists of a combination of waves with different heights, lengths and directions (known as the two-dimensional wave spectrum). Many ECMWF wave parameters (such as the mean wave period) give information averaged over all wave frequencies and directions, so do not give any information about the distribution of wave energy across frequencies and directions. This parameter gives more information about the nature of the two-dimensional wave spectrum, and represents a measure of the range of wave directions for each frequency integrated across the two-dimensional spectrum. It takes values between 0 and $\sqrt{2} \approx 1.4$. Where 0 corresponds to a unidirectional spectrum and $\sqrt{2}$ indicates a uniform spectrum (i.e., all wave frequencies coming from a different direction).*

*According to the Figure 11, this parameter is mainly distributed between 0.4 and 0.7 for considered freak wave events. This suggests, that crossing seas regime should not play a major role for the considered freak wave data.*

[Figure]

**Figure 10.** *The parameter kh vs the Benjamin Feir Instability index (BFI)*

[Figure]

**Figure 11.** *The parameter kh vs the wave spectral directional width*

A correlation between freak events and wind speed is presented. Waves are not locally generated but their amplitude is better correlated to the fetch (in time and space).

*Answer: Of course, the comment of the reviewer is absolutely correct. But from the reanalysis, we cannot recover many wave and wind parameters, and the correlation is determined by the roughest parameters. A good correlation confirms the connection of events with wind speed, and any deviations can be attributed to other wind and wave characteristics.*

No formula or discussion is presented for the gusts, but its obvious that if the formula is linear with respect to the wind speed, the correlation would be the same as for the wind speed. Moreover, correlations in deep and shallow water seem identical. Error bar on the coefficients should have been added and R2 value reported.

*Answer: We have also added the following clarification in the text:*
*Line 228: "Wind gust is the maximum wind gust at the specified time, at a height of ten meters above the Earth surface. It is defined as the maximum of the wind averaged over 3 second intervals. This duration is shorter than a model time step, and so the ECMWF Integrated Forecasting System (IFS) deduces the magnitude of a gust within each time step from the time-step-averaged surface stress, surface friction, wind shear and stability. Care should be taken when comparing model parameters with observations, because observations are often local to a particular point in space and time, rather than representing averages over a model grid box."*

*The values of R2 are given in the end of Section 3: "The coefficients of determination for both wind speed and gusts data for coastal events are around 0.5…. Dependence of wind speed and gusts versus significant wave heights for shallow and deep freak wave events and their linear approximations are presented in Fig. 12. The coefficients of determination for both wind speed and gusts data in this case are 0.68, which is larger than for coastal events."*

**Technical Comments**

First line of the abstract a definition of rogue waves is provided and the authors state that "and cause human loss…" but this should be "and can cause …". A definition of rogue waves is provided later on as twice the significant wave height, not all these waves have caused damage or loss. Only the reported ones.
*Answer: Agree. "Can" is added.*

The definition of the significant wave height, particularly in spectral wave models, is given as 4 times the square root of the zero-th order moment of the spectrum. The definition based on the mean of the highest third is now a bit outdated, and often substituted by the 4 times the standard deviation of the surface elevation.
*Answer: Corrected. Line 29: "It can also be defined as four times the standard deviation of the surface elevation, this definition is often used in spectral wave models (Massel, 1996; Kharif et al., 2009). The difference in magnitude between the two definitions is often only a few percent.")*

*Line 87: "Here we use the significant height of combined wind waves and swell ($H_s$) taken from the data of reanalysis, which is calculated as four times the square root of the integral over all directions and all frequencies of the two-dimensional wave spectrum."*

Dates are not consistent across the manuscript.
*Answer: Corrected.*

References are needed (and quotation marks as well) when quoting media and reports [around line 70 and 100].

*Answer: All references to media sources used in the text were added to the Reference list. The descriptions of the events were reformulated for not being the quoting media, therefore the quotation marks are not used.*

Line 132, why is an error of 10% accepted?

*Answer: One of the reasons for this error is that Hfr was replaced by Hmax since Hfr is unknown and can't be obtained from the model with the high precision. Hmax (the maximum individual wave height) is derived statistically from the two-dimensional wave spectrum. It can be considered as close value to Hfr but with certain error which we set as 10%. But, of course, this approach is not very accurate, since we are not talking about in-situ measurements. This clarification is added to the text (Line 155).*

I also want to note that ERA5 reanalysis performs poorly in coastal areas with complicated bathymetry. The resolution is approximately 1degree, and therefore does not accurately reproduce coastal areas.

*Answer: We agree. The following comment is added into the text. Line 237: "We should note that having a resolution of approximately 1 degree, ERA5 model does not perform well in coastal areas with complicated bathymetry."*

Individual waves can break well below the steepness 0.44, indeed sea states with a characteristic steepness of 0.12 have frequent breaking.

*Answer: The following comment is added into the text. Line 165: "However, individual waves can break well below the steepness 0.44. Indeed sea states with a characteristic steepness of 0.12 have frequent wave breaking."*

Line 141 and 142, dotes -> dots.
*Answer: Corrected*

The authors refer to the kurtosis, but the one they report is the excess kurtosis.
*Answer: "Kurtosis" is replaced by "excess kurtosis".*

The Conclusions do not provide any insight, critical comments. It reads a lot like a repetition of the previous sections.
*Answer: The Conclusion was shortened, and detailed descriptions of results which were given in the main text were removed.*

Figures with shading limit readability (e.g. Fig 5 and 6). Color palette choice is not color-blind friendly.
*Answer: The color palette of Fig.5 and Fig.6 has been changed.*

Figure 7 would have benefitted from lines at various steepness to immediately identify the steepness of the event.
*Answer: Several lines corresponding to steepnesses kH/2=0.44, kH/2=0.33, kH/2=0.22, kH/2=0.11 have been added.*